# Selenocysteine Machinery Primarily Supports TXNRD1 and GPX4 Functions and Together They Are Functionally Linked with SCD and PRDX6

**DOI:** 10.3390/biom12081049

**Published:** 2022-07-28

**Authors:** Didac Santesmasses, Vadim N. Gladyshev

**Affiliations:** Division of Genetics, Department of Medicine, Brigham and Women’s Hospital, Harvard Medical School, Boston, MA 02115, USA; dsantesmassesruiz@bwh.harvard.edu

**Keywords:** selenocysteine, selenium, selenoprotein, co-essentiality network

## Abstract

The human genome has 25 genes coding for selenocysteine (Sec)-containing proteins, whose synthesis is supported by specialized Sec machinery proteins. Here, we carried out an analysis of the co-essentiality network to identify functional partners of selenoproteins and Sec machinery. One outstanding cluster included all seven known Sec machinery proteins and two critical selenoproteins, GPX4 and TXNRD1. Additionally, these nine genes were further positively associated with PRDX6 and negatively with SCD, linking the latter two genes to the essential role of selenium. We analyzed the essentiality scores of gene knockouts in this cluster across one thousand cancer cell lines and found that Sec metabolism genes are strongly selective for a subset of primary tissues, suggesting that certain cancer cell lineages are particularly dependent on selenium. A separate outstanding cluster included selenophosphate synthetase SEPHS1, which was linked to a group of transcription factors, whereas the remaining selenoproteins were linked neither to these clusters nor among themselves. The data suggest that key components of Sec machinery have already been identified and that their primary role is to support the functions of GPX4 and TXNRD1, with further functional links to PRDX6 and SCD.

## 1. Introduction

Selenocysteine (Sec) is a rare proteinogenic amino acid analogous to cysteine (Cys) that contains selenium instead of sulfur. Sec is known as the twenty-first amino acid in the genetic code and is used in selenium-containing proteins, or selenoproteins. Selenium is an essential trace element for humans and other mammals, and selenoproteins mediate its main biological functions. Most selenoproteins are oxidoreductase enzymes with Sec being a critical residue for their function, as it is generally found at their catalytic redox-active sites. Selenoproteins serve diverse functions, which include maintenance of redox homeostasis, inhibition of ferroptosis, metabolism of thyroid hormones, and synthesis of selenophosphate, among others [1].

The synthesis of selenoproteins involves the biosynthesis and insertion of Sec, and requires a dedicated machinery [2]. Unlike the 20 canonical amino acids, Sec is synthesized on its own tRNA following a multi-step process that uses selenophosphate as the selenium donor [3]. Sec is then inserted co-translationally into the growing peptide in response to the UGA codon. The main signal for Sec insertion is the RNA structure in the 3′UTR called SECIS element [4]. Since UGA is normally a stop signal, recoding UGA codons to code for Sec is in competition with translation termination. Failed recoding gives rise to truncated proteins that lack the Sec residue and often leads to the degradation of the mRNA by nonsense-mediated decay (NMD) because UGA is read as a premature termination codon [5]. The synthesis of full-length selenoproteins requires the complete Sec machinery and is highly dependent on the availability of selenium [6]. Inactivating mutations in the synthesis machinery factors lead to congenital disorders caused by global selenoprotein deficiency [7].

Co-essentiality analyses have the potential to uncover functional links between proteins. The approach is based on measuring the fitness of single-gene perturbations across multiple conditions and mapping functional interactions by correlating the resulting phenotype. A co-essentiality network was recently developed by applying this method to a genome-wide CRISPR screening across 485 cancer cell lines with gene-level essentiality scores [8]. Essentially, the idea is that two genes with a very similar impact on cell growth across hundreds of cell lines in response to knockdown (siRNA) or knockout (CRISPR) are expected to be functionally linked. Such paired functional links are then integrated into a global network, revealing clusters of functionally linked proteins. This network could recover co-essential gene modules that recapitulate well-known pathways, and allow novel functions to be assigned to some uncharacterized proteins [8].

The human genome encodes 25 selenoproteins and 7 well-established and several additional candidate Sec machinery proteins, as well as one Sec-specific tRNA [9]. We analyzed here the co-essentiality network to identify the functional partners of selenoproteins and proteins comprising the Sec machinery. We found a remarkable cluster that included all seven known genes required for the biosynthesis and insertion of Sec, two critical selenoproteins, GPX4 and TXNRD1, and two additional proteins, PRDX6 and SCD. We also analyzed the essentiality scores of knockouts of the proteins in this cluster across one thousand cancer cell lines and found that the proteins involved in the metabolism of Sec are strongly selective for a subset of primary tissues, suggesting that specific cancer cell lineages were particularly dependent on selenium. An additional well-defined cluster included selenophosphate synthetase SEPHS1, which is not involved in Sec machinery. We discuss how the co-essentiality network can identify new functional partners, thereby clarifying the role of selenium in biology and medicine.

## 2. Materials and Methods

To identify gene clusters and modules linked to Se metabolism, we used the co-essentiality network (http://coessentiality.net/; accessed on 16 May 2022) [8]. The tool offers a genome-wide 2D map based on the co-essentiality of genes in 485 human cancer cell lines from the Cancer Dependency Map [10]. The dataset links genes by proximity in the network and performs enrichment analyses. The coordinates of the 2D network map were downloaded from https://github.com/kundajelab/coessentiality/blob/master/vizdf.tsv (accessed on 16 May 2022).

We further employed the Cancer Dependency Map (DepMap) portal (https://depmap.org/portal/; accessed on 16 May 2022) and used several datasets representing each of the 941 cancer cell lines included in the release 22Q1. We also utilized the CRISPR gene effect dataset (CRISPR_gene_effect.csv; release 22Q1), which contains cell viability scores from the Achilles project [10], a genome-wide CRISPR screen of genetic perturbations to silence or knockout individual genes and identify those genes that affect cell survival. We used the metric gene effect (or cell viability), whereby a lower score means that a gene is more likely to be dependent on a given cell line. A score of 0 is equivalent to a gene that is not essential whereas a score of −1 corresponds to the median of all common essential genes. We computed the pairwise Pearson’s correlation coefficients of cell viability between our genes of interest across all available cancer cell lines and plotted them as heatmaps for visualization or pairwise scatter plots. We also analyzed gene expression TPM values of the protein-coding genes for DepMap cell lines (CCLE_expression.csv; release 22Q1). Values contained in this file were inferred from RNA-seq data using the RSEM tool and were reported after log2 transformation, using a pseudo-count of 1; log2(TPM + 1). We then generated the distribution of gene expression TPM and average gene expression TPM across primary tissues in hierarchical clustered heatmaps. Hierarchical clustering of both columns and rows in the heatmaps was performed using the following clustering parameters: clustering_distance_cols=“euclidean”, clustering_distance_rows=“euclidean”, clustering_method=“ward.D” using the pheatmap function from the pheatmap package (by Raivo Kolde; https://github.com/raivokolde/pheatmap, accessed on 16 May 2022) version 1.0.12 for R.

The STRING database [11] was used to explore the functional interactions of PRDX6 (https://string-db.org; accessed on 16 May 2022). The following parameters were used to produce Figure 1e: Network type = Full; network edges = confidence; interaction sources = Experiments + Textmining; minimum required interaction score = 0.4; max number of interactors to show = 1st shell: 10 interactions and 2nd shell: none.

Among our genes of interest, two selenoproteins were missing in the Cancer Dependency Map dataset, SELENOF and TXNRD3; thus, these could not be analyzed here. Although this is a limitation of this bioinformatics approach, the functions of SELENOF and TXNRD3 and their roles in disease have been experimentally analyzed recently by our group and others [12,13,14,15].

## 3. Results

### 3.1. Co-Essentiality Network

Selenoproteins serve various functions, are expressed in different cell compartments, and exhibit tissue-specific patterns (e.g., TXNRD3, SELENOV, GPX6, GPX3, DIO1, DIO2, DIO3). Co-essentiality analyses have an opportunity to uncover novel functional links involving selenoproteins and Sec machinery genes. We have analyzed the co-essentiality network built from gene essentiality scores across 485 human cancer cell lines, to identify the coordinates of all selenoproteins. The 23 selenoproteins included in the network were generally scattered in the 2D map, except for two selenoproteins (Figure 1a). The groups of genes clustered with individual selenoproteins are included in Appendix A. Interestingly, well-established Sec machinery genes clustered together, whereas candidate Sec machinery proteins TRNAU1AP and SECISBP2L clustered neither with selenoproteins nor with established Sec machinery components.

### 3.2. Selenocysteine Gene Module

We identified a remarkable, very tight functional cluster of 11 proteins, which is well separated from other proteins in the network (Figure 1b). It includes all 7 genes known to be involved in Sec biosynthesis and insertion, i.e., LRP8 (also known as ApoER2, a receptor for SELENOP (Selenoprotein P), which delivers Se in the form of Sec to target tissues) [16], SCLY (Sec lyase that degrades Sec for the use of selenium in other selenoproteins) [17], SEPHS2 (selenophosphate synthetase 2 that forms selenophosphate from selenide and ATP) [18], PSTK (phosphoseryl-tRNASec kinase that prepares the serylated Sec tRNA for Sec biosynthesis) [19], SEPSECS (Sec synthetase that selenylates phospho-serine on Sec tRNA) [20,21], SECISBP2 (selenocysteine insertion sequence-binding protein 2 that is required for Sec insertion into proteins) [22] and EEFSEC (Sec-specific elongation factor that inserts Sec into nascent polypeptide chains) [23]. Pathway enrichment analysis of this cluster of proteins showed a highly significant enrichment of selenium metabolism pathways (Figure 1c). This is the first time that a functional clustering of all these Sec machinery proteins is observed. This finding supports the idea that all specialized protein components of the selenoprotein synthesis pathway had been already identified.

In addition to the 7 proteins known to be involved in Sec biosynthesis and insertion, this cluster includes the most critical selenoproteins, GPX4 (glutathione peroxidase 4) and TXNRD1 (thioredoxin reductase 1), whereas all other selenoproteins do not show functional links to this cluster or among themselves (Figure 1f and Appendix A).

The tenth protein in the cluster is SCD (stearoyl-CoA desaturase), but in contrast to all other proteins in the cluster, this protein is negatively correlated with Sec biosynthesis and selenoproteins, particularly with GPX4 (Figure 1d and Appendix A).

The last and perhaps unexpected protein in the cluster is PRDX6. This is peroxiredoxin 6, an enigmatic thiol oxidoreductase [24,25]. Our co-essentiality analysis clearly links PRDX6 to the Sec pathway (Figure 1b), but this protein is not a selenoprotein. There are several possible functions for this protein in the pathway and they are discussed in detail in the discussion section.

### 3.3. Selenium-Dependent Cancer Cell Lines

The Achilles project is a genome-wide CRISPR screen of genetic perturbations on cancer cell lines [10]. We analyzed cell viability scores of gene knockouts of proteins in the Sec cluster described above, as well as the rest of selenoproteins (Figure 2a). Among the genes analyzed, the strongest effect on cell viability was observed for GPX4 (Figure 2a). GPX4 is known to be essential for human cells in culture. It was observed that cells lacking the Sec tRNA are viable only in a GPX4-Cys background, that is if Sec in GPX4 is replaced with Cys and therefore the GPX4 activity does not depend on selenium [26]. Accordingly, the genes necessary for the synthesis of selenoproteins also showed strong gene effects, including PSTK, SEPSECS, the selenoprotein SEPHS2, and EEFSEC (Figure 2a). On the other hand, SECISBP2, the SECIS binding protein, did not show a negative effect on viability (Figure 2a). This could be explained by the previous observation that some level of UGA readthrough might occur in the absence of SECISBP2 [27], which could lead to some level of selenoprotein synthesis. It is also possible that a partial SECISBP2 function could be retained in these knockout assays, as it has been proposed in some patients with potential alternative start sites [7]. Another possibility could be that SECISBP2L, the SECISBP2 paralog, might compensate for its deficiency, which has been hypothesized previously [28,29]. SECISBP2L, however, did not cluster with the selenoprotein synthesis genes in the co-essentiality network (Figure 1a). Nevertheless, a more informative assay would knockout SECISBP2L in a SECISBP2 null background, and then the cell viability score would be assessed.

We observed that the distribution of cell viability scores for GPX4 was bimodal, suggesting that there were two populations of cells distinguished by their dependency on this protein (Figure 2b). To further analyze this, we broke down the cell lines into lineages based on primary tissue of origin (Appendix A). We observed that a subset of primary tissues had lower viability scores for GPX4 and the other proteins in the Sec cluster, including the selenoprotein TXNRD1 and the Sec machinery factors SEPSECS, EEFSEC, and SEPHS2 (Figure 2c). The primary tissues that appeared to be most dependent on selenoproteins and selenium included the thyroid, kidney, liver, blood, and central and peripheral nervous systems. On the other hand, primary tissues that appeared to be less dependent on selenium included several organs in the digestive system (upper aerodigestive, esophagus, gastric, bile duct, pancreas, and colorectal), cervix, breast, lung, skin, prostate, and urinary tract (Figure 2c). In order to assess whether the observed differences in cell viability across lineages were caused by changes in gene expression, we analyzed RNA-seq performed on the same cancer cell lines. The top expressed selenoprotein was GPX4, but in this case, the distribution was not bimodal (Figure 3a). In contrast to cell viability, we did not observe differences in gene expression across lineages (Figure 3b). Indeed, we did not observe a correlation between cell viability and gene expression for the genes in the Sec cluster (Figure 3c), which suggests that the differences in cell viability across lineages observed in genes in the Sec cluster (Figure 2c) were not caused by changes in the expression of individual genes (Figure 3c).

### 3.4. Selenophosphate Synthetase 1 Gene Module

Another interesting functional cluster involves 15 proteins. One of those proteins is SEPHS1 (Figure 4a). This cluster was previously described but no mention of the possible involvement of SEPHS1 was discussed [8]. Interestingly, almost all proteins in this cluster are transcription factors, which are uniquely essential in breast cancer cell lines [8]. SEPHS1 is the paralog of selenophosphate synthetase SEPHS2, an essential selenoprotein in the Sec synthesis pathway that is part of the Sec gene module described above. Our previous studies revealed that SEPHS1 evolved by gene duplication of SEPHS2 during metazoan evolution [30]. SEPHS1 and SEPHS2 are both present in vertebrates and insects, but more ancestral species have just one ancestral protein. SEPHS1 is not a selenoprotein. This protein carries, instead of Sec, threonine in vertebrates and arginine in insects [30]. It is possible that SEPHS1 supports a function similar to selenophosphate synthesis but unrelated to Sec biosynthesis. Indeed, we found that while SEPHS1 is essential for mammalian development, it is also found in insects that do not have selenoproteins, and that it does not participate in the selenoprotein synthesis [31,32].

SEPHS1 clusters with a breast cancer-specific gene module that contains ESR1, the estrogen receptor (ER), which is overexpressed in breast cancers and enables hormone-dependent growth, and several other genes that functionally interact with ESR1, including transcriptional regulators in ER-dependent breast cancers SPDEF, FOXA1, and GATA3 [8].

## 4. Discussion

Selenium is an essential trace element incorporated into selenoproteins in the form of the twenty-first amino acid Sec. Selenium is obtained by organisms from food and is distributed throughout the body to peripheral tissues, where it is used to synthesize selenoproteins. Functionally characterized selenoproteins are oxidoreductase enzymes that mediate the main biological functions of selenium in humans. The synthesis of selenoproteins is a complex process that involves the uptake of selenium by cells, biosynthesis of Sec, and its insertion into proteins in response to a UGA codon. The whole process of selenoprotein synthesis requires a dedicated Sec machinery.

Since the discovery of the presence of selenium in proteins, the elucidation of the pathway for the synthesis of selenoproteins took place over many years based on experiments carried out by various laboratories. The biosynthesis of Sec takes place on its own Sec tRNA in a multi-step process. The Sec tRNA is initially aminoacylated with serine (Ser) by seryl tRNA synthetase (SARS). In eukaryotes, Ser is then phosphorylated by phosphoseryl-tRNASec kinase (PSTK). The last step in Sec synthesis is selenylation of phosphoserine, performed by the Sec synthase (SEPSECS). Sec is then inserted co-translationally during the translation of selenoprotein mRNAs into the growing peptide in response to a UGA codon [1]. UGA codons normally signal a translational stop, and Sec insertion involves recoding UGA to specify Sec rather than to stop translation. This process occurs only in selenoprotein mRNAs and is mediated by the SECIS element, an RNA stem loop in the 3′UTR. The factors required for Sec insertion are the SECIS binding protein SECISBP2, and the Sec-specific elongation factor EEFSEC. Additional factors have been proposed to be involved on higher levels of selenoprotein expression regulation [33]. The selenoprotein synthesis machinery factors have been discovered or validated by their co-occurrence with selenoproteins in sequenced genomes using bioinformatics analyses. However, the overall composition of the Sec machinery remained unclear. Remarkably, we identified a gene module in a co-essentiality network that contains all seven known specialized proteins required for Sec biosynthesis and insertion. This is the first time that all these Sec machinery factors showed functional clustering. This finding supports the idea that all such factors had already been identified.

The distribution of hepatic selenium throughout the body and its supply to peripheral tissues is carried out by selenoprotein P (SELENOP), a unique selenoprotein that contains multiple Sec residues in its C-terminal domain. SELENOP is an extracellular selenoprotein mainly produced in the liver and abundantly found in plasma [34]. Extrahepatic tissues acquire hepatic selenium primarily by endocytosis of SELENOP mediated by LRP8 (also known as ApoER2), an endocytic receptor that participates in the cell-specific SELENOP uptake and retention [35]. SELENOP is also known to be taken up by another receptor, megalin, and by pinocytosis [34,36]. Selenium is recycled within cells to maintain the synthesis of selenoproteins. Sec in SELENOP and in other selenoproteins is freed by proteolytic degradation. Selenocysteine lyase (SCLY) is the specific enzyme for metabolizing free Sec to selenide and alanine, providing selenium for the synthesis of new selenoproteins [17]. Selenide then re-enters Sec synthesis by binding to phosphate to generate selenophosphate by selenophosphate synthetase SEPHS2 [37]. Some tissues, particularly the brain and testes, use the SELENOP cycle to preserve tissue selenium in conditions of poor selenium supply. However, SELENOP was not part of the Sec metabolism cluster. It is possible that the importance of the endogenous SELENOP might have been artificially lowered on these assays due to the presence of SELENOP in the media where cells were cultured.

The dependency on selenium has been described as a liability for cancer cells and proteins LRP8 and SEPHS2, which are involved in the metabolism of Sec, have been proposed as potential drug targets for cancer. LRP8 is required for selenium uptake in cells and promotes resistance to ferroptosis in cancer cells due to the expression of GPX4 [38]. SEPHS2 is responsible for the synthesis of selenophosphate from selenide and ATP. Selenophosphate is the selenium donor required for GPX4 expression. By doing so, SEPHS2 detoxifies selenide, which is toxic for cells [39]. The gene knockout essentiality scores analyzed here showed that a subset of cancer lineages (primary tissues) might be more susceptible to selenium deficiency, as genes involved in Sec metabolism are strongly selective (Figure 2). Cancer cell lineages such as liver, kidney, and blood appeared to be most dependent on selenium. It is important to note, however, that the co-essentiality network is based on cell viability scores derived from cell cultures, whereas physiological conditions might be very different. For example, the media used in cell cultures usually contains SELENOP. It is possible that LRP8 is essential in cell culture because SELENOP, which is obtained from the media, is the only source of selenium [38].

In addition to selenoprotein synthesis factors, the Sec gene module also includes two critical selenoproteins, GPX4 and TXNRD1. The functional link between the Sec machinery factors and these selenoproteins is the use of Sec, i.e., Sec is needed in cells when GPX4 and TXNRD1 are needed. GPX4 protects cells from ferroptosis, a recently described form of cell death that is accompanied by a large amount of iron accumulation and lipid peroxidation [40,41]. GPX4 is a member of glutathione peroxidases, a large protein family that includes 8 proteins in mammals, with five of them being selenoproteins [42]. GPX4 stands out among the Sec-containing glutathione peroxidases because of its ability to reduce lipid hydroperoxides and the use of protein thiols as donors of electrons in addition to the glutathione [43]. GPX4 has a master role in a recently described form of non-apoptotic cell death called ferroptosis [40,41], and has been linked to the indispensable role of selenium in cell viability [26]. TXNRD1 is an essential component of the thioredoxin system, a major disulfide reducing system that modulates multiple cellular processes such as antioxidant response, apoptosis, and cell proliferation. The main physiological role of TXNRD1 is the NADPH-dependent reduction of thioredoxin 1 (TXN) [44]. The TXN system provides reducing equivalents to ribonucleotide reductase, which is responsible for the synthesis of 2′-deoxyribonucleotides. Studies in mice showed impaired pyrimidine and purine biosynthesis after T-cell stimulation in the absence of Txnrd1 [45], which could explain the TXNRD1 essentiality.

PRDX6 was also part of the Sec cluster. PRDX6 has several enzymatic activities: peroxidase, phospholipase A2 (PLA2), and acyl transferase. Its gene has been associated with neuropsychiatric disorders [46]. PRDX6 is an unusual peroxiredoxin because it has a single catalytic Cys, and can reduce phospholipid hydroperoxides [47], which suggests that it is important for the repair of damage to lipids in the cell membrane. The only other enzyme in mammalian cells that can reduce phospholipid hydroperoxides is GPX4. Studies on *Prdx6* knockout mice showed a deficiency in phospholipid catabolism and higher oxidation levels for lipids and proteins. Interestingly, no detectable enzymatic activity for the reduction of phospholipid hydroperoxides was observed in the lungs, suggesting that the lungs of these mice did not contain GPX4 [25]. First, PRDX6 may support GPX4 function through heterodimerization (both proteins are thioredoxin-fold proteins, and dimerization, tetramerization, and dodecamerization are often observed among such proteins [47,48,49,50,51,52]). GPX4 is known as a monomeric glutathione peroxidase, whereas most other glutathione peroxidases are tetramers. GPX4 is also known to interact with other proteins, e.g., it moonlights into a structural protein that multimerizes and crosslinks with other proteins during sperm maturation [48]. Moreover, protein–protein interaction databases and yeast two hybrid datasets have identified an association between GPX4 and PRDX6 (Figure 1e). It should be stressed that GPX4 and PRDX6 are positively correlated in the co-essentiality network (Figure 1d and Appendix A), which means that the cells need GPX4 when they need PRDX6 and vice versa. Therefore, it is unlikely that PRDX6 functions as an alternative phospholipid hydroperoxide peroxidase. It is possible that PRDX6 modifies the GPX4 activity for targeting different forms of phospholipid hydroperoxides or that the heterodimers reduce diverse substrates compared to GPX4 alone. While we consider the heterodimerization hypothesis more likely, there are alternative possibilities. One is that PRDX6 is involved in the use of SELENOP-derived Sec for selenoprotein biosynthesis. For example, it may form selenenyl-sulfides with such Sec in peptides derived from SELENOP, thereby trapping Sec for subsequent use of selenium. It should be noted that PRDX6 is functionally related to other peroxiredoxins, which are oxidized by peroxides and oxidize thiols or selenols in proteins (or in glutathione). Thus, the function to support GPX4 activity, as well as the function of oxidizing Sec, may be consistent with its general functionality. A third possibility is that its functional association with the Sec cluster is due to its reduction being fueled by TXNRD1 via TXN (thioredoxin 1), i.e., TXNRD1 reduces TXN, which in turn reduces PRDX6. Further experimental analyses are required to test these possibilities to define the function of PRDX6 and explain the functional link of this thiol oxidoreductase to the Sec pathway.

The last protein in the Sec cluster was SCD (stearoyl-CoA desaturase). SCD is known to contribute to phospholipid hydroperoxide reduction [53], a function that overlaps with that of GPX4. This is consistent with the presence of SCD in the cluster and suggests that SCD could compensate for the decreased levels of GPX4 or Sec machinery components.

A partial Sec cluster had been reported before, in an earlier co-essentiality network built using an earlier version of the Cancer Dependency Map, which included fewer cell lines [54]. The incomplete cluster was missing proteins LRP8, SCLY, SECISBP2, and TXNRD1. These proteins are known to be directly related to Sec, and the fact that they were recovered in the co-essentiality network analyzed here, which was built from 485 cancer cell lines, underlines the importance of gain in statistical power by a larger sample size. Moreover, the proteins SCD and PRDX6, which were identified here as candidate functional partners of selenoproteins, were also missing in the previous network.

It is important to mention that AIMF2 (also known as FSP1, the ferroptosis suppressor protein 1) did not cluster with GPX4, despite being a known functional partner of GPX4 [55,56]. No strong correlation coefficient of cell viability scores was observed between AIMF2 and GPX4, and other proteins in the Sec cluster (DepMap portal). Another candidate protein proposed to be related to selenium is the selenium-binding protein 1 (SELENBP1). This protein does not contain Sec, and its mode of hypothetically binding Se is unclear. The function of this protein is currently unknown but might be implicated in cancer and other diseases [57]. We found that SELENBP1 was not part of the Sec cluster and did not correlate with the Sec machinery or other selenoproteins in terms of cell viability.

We also described a cluster involving SEPHS1. This protein was discovered as selenophosphate synthetases 1, but the evidence supports that it is not involved in selenoprotein synthesis. Whether its function is related to a novel use of selenium is not known. It is interesting, however, that it forms such a prominent and well-defined cluster with several transcription factors (Figure 4). Its presence in this cluster suggests that it may be required for the function of other proteins in the cluster. One possibility is that SEPHS1 phosphorylates thiols in these proteins (by analogy to phosphorylation of selenium by SEPHS2). While the specific function of SEPHS1 remains unknown, the newly discovered functional linkage provides molecular targets for future functional analyses. Additional experiments will be required to characterize its function and unravel its relationship with other proteins in the cluster.

The main biological functions of selenium in mammals are mediated by selenoproteins, but the functions of many selenoproteins remain unknown. We propose that co-essentiality networks have the potential to uncover functional protein partners of selenoproteins that can shed light on the functions of uncharacterized selenoproteins and the pathways they are involved in.

## Figures and Tables

**Figure 1 biomolecules-12-01049-f001:**
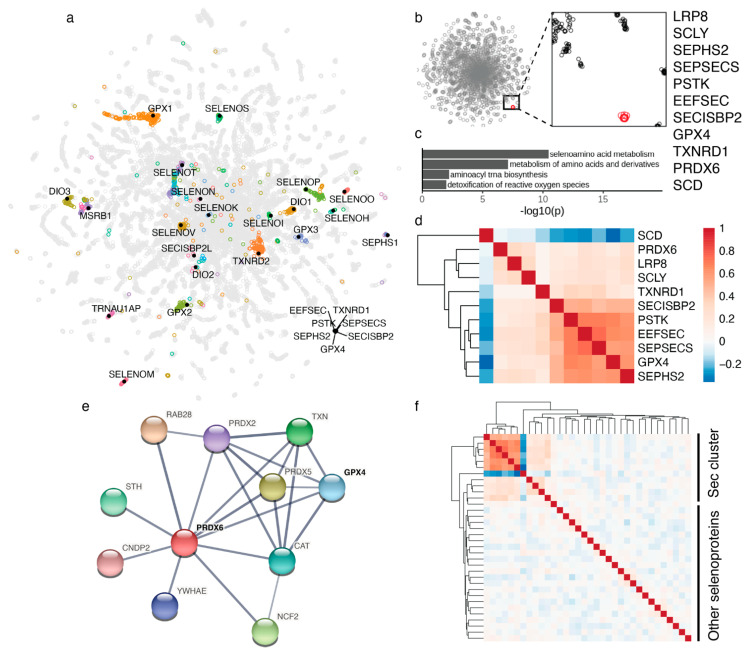
Selenocysteine gene module in the co-essentiality network. (**a**) 2D map of the genome-wide co-essentiality network showing the location of selenoproteins and Sec machinery proteins. Other proteins clustered with individual selenoproteins are highlighted by different colors. (**b**) Co-essentiality network with an expanded view showing tight clustering of 11 proteins (shown in red) in the Sec cluster. (**c**) Pathway enrichment analysis of proteins in the Sec cluster. (**d**) Correlation heatmap of cell viability scores between 11 proteins in the Sec cluster. Values correspond to Pearson’s correlation across 941 cancer cell lines from the Cancer Dependency Map [10]. (**e**) Interaction network based on experimental evidence and text mining from STRING database [11] showing a functional connection between PRDX6 and GPX4. (**f**) Correlation heatmap of cell viability scores between all proteins in the Sec cluster and the rest of selenoproteins. The top left corner corresponds to the same proteins as in panel (**d**). A larger version of the heatmap including protein names is provided as Appendix A.

**Figure 2 biomolecules-12-01049-f002:**
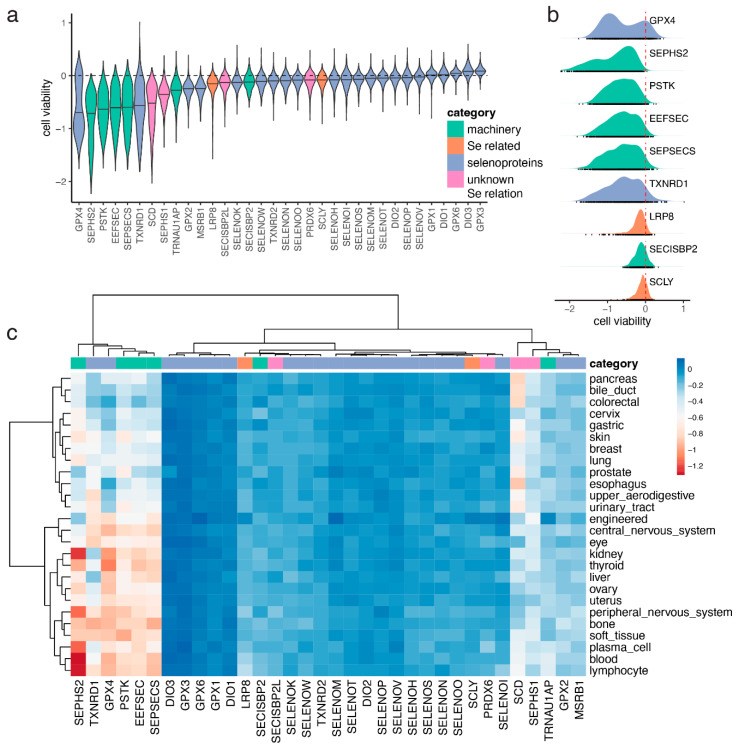
Cell viability of gene knockouts across cancer cell lines. (**a**) Distribution of cell viability scores for proteins in the Sec cluster and the rest of selenoproteins. (**b**) Density plot of cell viability scores for the indicated genes. (**c**) Heatmap showing the average cell viability scores for the same genes across cell lineages (primary tissues). Proteins are colored based in their category. All panels follow the legend in panel a. Categories include Sec machinery proteins, selenoproteins (not including the selenoprotein SEPHS2 which is part of the machinery), selenium-related proteins (LRP8 and SCLY), and proteins with unknown relationship with selenium or selenoproteins.

**Figure 3 biomolecules-12-01049-f003:**
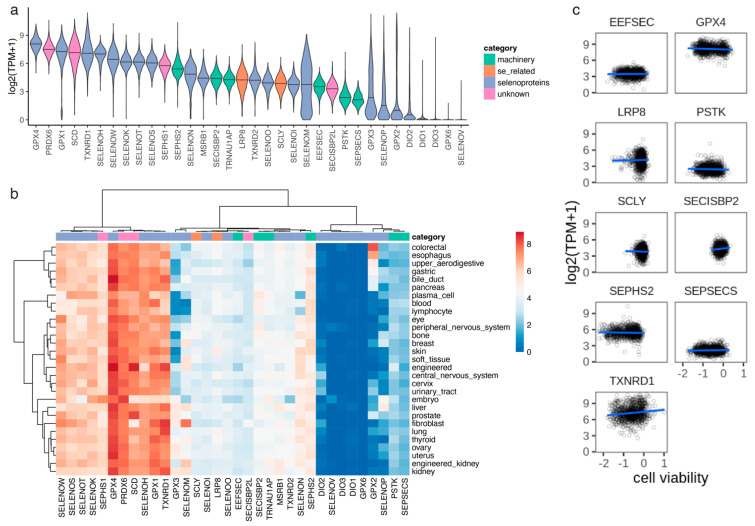
Gene expression measured by RNA-seq in cancer cell lines. (**a**) Distribution of gene expression (log2(TPM+1)) across 941 cancer cell lines. (**b**) Average gene expression (median log2(TPM+1)) across cancer lineages (primary tissues). (**c**) Linear regression model between cell viability and gene expression for proteins in the Sec cluster.

**Figure 4 biomolecules-12-01049-f004:**
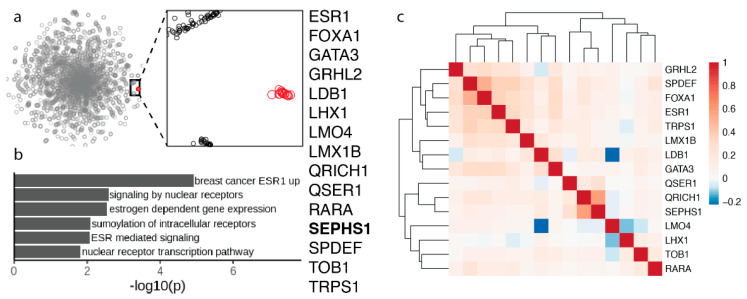
Selenophosphate synthetase 1 gene module. (**a**) 2D map of the genome-wide co-essentiality network showing the cluster in red. The expanded view shows the tight clustering of the 15 proteins (shown in red) in the cluster. (**b**) Pathway enrichment analysis for the proteins in the cluster, which are listed on the right. (**c**) Correlation heatmap of cell viability scores between the 15 proteins in the cluster. The values correspond to Pearson’s correlation across 941 cancer cell lines from the Cancer Dependency Map [10].

## Data Availability

Not applicable.

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
