# Peer review of "Selenocysteine Machinery Primarily Supports TXNRD1 and GPX4 Functions and Together They Are Functionally Linked with SCD and PRDX6"

_biomolecules, 2022, doi:10.3390/biom12081049_

Round 1

Reviewer 1 Report

The authors performed data mining on co-essentiality data across cell lines and discovered that two selenoproteins and all known members of the core selenoprotein biosynthesis machinery clustered together. They found that SCD and PRDX6 were part of this cluster supporting previous hints that PRDX6 is functionally linked to selenoproteins. Another important finding was that factors often included among the selenoprotein biosynthesis factors like SECISBP2L and TRNAU1AP (and SEPHS1) are not part of this cluster.

Not surprisingly, gene expression levels and essentiality did not correlate, but it is important to see this data, since there are too many authors deducing importance from mRNA levels.

The authors found that SEPHS1 did not cluster with selenoproteins or the biosynthesis machinery, but rather clustered with a number of trancriptional regulators important in estrogen-dependent breast cancer cell lines. This may be an important hint to finally uncover the elusive function of this protein!

This reviewer is very happy with the paper as it is, except:

- line 65: must read PRDX6

- it would be nice, if the authors commented on FSP1/AIMP2, which, interestingly, does not cluster with GPX4 despite a functional link

- SELENBP1 is sometimes functionally linked to selenoproteins. It does not cluster with these candidates. This reviewer knows that its nature as a secreted protein may be the reason for the lack of correlation, but maybe a short comment would be nice.

- lines 2004- need adjustment

- more of a request: The authors do not comment a lot on what is known about PRDX6. It may be nice to mention its different catalytic functions and mention e.g. mouse KO models (see Molecular Neurobiology (2021) 58:4348–4364)

- line 310: "misacylated" seems a bit strong given that throughout evolution there is a consensus that tRNA(Sec) is aminoacylated with Ser.

Author Response

The authors performed data mining on co-essentiality data across cell lines and discovered that two selenoproteins and all known members of the core selenoprotein biosynthesis machinery clustered together. They found that SCD and PRDX6 were part of this cluster supporting previous hints that PRDX6 is functionally linked to selenoproteins. Another important finding was that factors often included among the selenoprotein biosynthesis factors like SECISBP2L and TRNAU1AP (and SEPHS1) are not part of this cluster.

Not surprisingly, gene expression levels and essentiality did not correlate, but it is important to see this data, since there are too many authors deducing importance from mRNA levels.

The authors found that SEPHS1 did not cluster with selenoproteins or the biosynthesis machinery, but rather clustered with a number of trancriptional regulators important in estrogen-dependent breast cancer cell lines. This may be an important hint to finally uncover the elusive function of this protein!

We thank the reviewer for the critical revision of our work and appreciate the positive comments. We have addressed all the concerns, and we have replied to each of them below.

This reviewer is very happy with the paper as it is, except:

- line 65: must read PRDX6

Thank you for catching this mistake, we have changed PRDX4 to PRDX6

- it would be nice, if the authors commented on FSP1/AIMP2, which, interestingly, does not cluster with GPX4 despite a functional link

Thank you for the relevant suggestion. We have added a sentence in the discussion (line 739) to point out this observation. We found that no proteins in the Sec cluster were among the top genes correlated with AIFM2 in terms of cell viability. We have also added two references that describe the functional link between AIMP2 and GPX4. We referred to the protein as AIFM2 in the text because, although it has been proposed renamed it as FSP1, there is another unrelated protein named FSP1 (atlastin GTPase 1) in the annotation used in the co-essentiality browser, so it would be less confusing for the reader.

- SELENBP1 is sometimes functionally linked to selenoproteins. It does not cluster with these candidates. This reviewer knows that its nature as a secreted protein may be the reason for the lack of correlation, but maybe a short comment would be nice.

Thank you for this suggestion. We have added a sentence in line 746 to include this observation.

- lines 2004- need adjustment

Thank you, we have adjusted the lines.

- more of a request: The authors do not comment a lot on what is known about PRDX6. It may be nice to mention its different catalytic functions and mention e.g. mouse KO models (see Molecular Neurobiology (2021) 58:4348–4364)

Thank you for this relevant request. We have added additional information on the functions of PRDX6 and KO models, as well as references.

- line 310: "misacylated" seems a bit strong given that throughout evolution there is a consensus that tRNA(Sec) is aminoacylated with Ser.

Thank you for the suggestion, we have changed misacylated to aminoacylated.

Reviewer 2 Report

This manuscript from Santemasses and Gladyshev is an excellent demonstration of the power of publicly available data.  It reveals novel functionality that, while still requiring molecular validation, may have never been brought to light without this kind of field-specific analysis.  Some minor comments:

1. Some strictly introductory material appears in the Discussion and some discussion - like material appears in the results. I would recommend expanding the intro and combining results and discussion

2. Line 66 "involve" should be involved

3. Line 76 "allows to link" is awkward.  consider changing to "links"

4. Line 125.  It is a bit misleading to say "Sec lyase that degrades Sec in Selenoprotein P...." as it does not specifically target those Sec residues.  If it were required to release Se from SELENOP, then the SCLY KO mouse would phenocopy the SELENLOP KO mouse and it does not.

5. Line 126 Why not use the SELENOP abbreviation as for other selenoproteins.

6. Line 159.  I might disagree that the foundation of TXNRD1 essentiality lies in "potent antioxidant cellular defense..."  Isn't it more likely related to is relationship to dNTP synthesis?

7. Line 196. The sentence "Thus, like the funcion..." is pretty unclear - maybe rephrase?

8. Line 213.  If I understand how the coessentiality works, then SECISBP2L should not sort with the essential factors because it has been demonstrated to be nonessential (KO has no effect on selenoprotein production unless SECISBP2 is impaired).  It would only sort as such in a SBP2 null background, correct?

9. Line 239 "in" is repeated

10. Line 311.  I don't think it is correct to call tRNASec "misacylated" by the SerRS, which implies a mistake.

Line 347.  Not only is it possible that LRP* essentiality is an artifact of cell culture, it was clearly demonstrated in Li et al. (NCB).  Along those lines, it may be important to note that SELENOP can also be taken up by other means (Megalin and pinocytosis).

Author Response

This manuscript from Santemasses and Gladyshev is an excellent demonstration of the power of publicly available data.  It reveals novel functionality that, while still requiring molecular validation, may have never been brought to light without this kind of field-specific analysis.  Some minor comments:

We thank the reviewer for the critical review of our work. We have addressed the concerns in the manuscript and replied to each of them below.

  1. Some strictly introductory material appears in the Discussion and some discussion - like material appears in the results. I would recommend expanding the intro and combining results and discussion

Thank you, we have moved some parts from results to discussion, as we agree it would help to improve the structure of the manuscript.

  1. Line 66 "involve" should be involved

Thank you, the typo it has been corrected.

  1. Line 76 "allows to link" is awkward. consider changing to "links"

Thank you, we agree with this suggestion, and we have now changed it.

  1. Line 125. It is a bit misleading to say "Sec lyase that degrades Sec in Selenoprotein P...." as it does not specifically target those Sec residues. If it were required to release Se from SELENOP, then the SCLY KO mouse would phenocopy the SELENLOP KO mouse and it does not.

Thank you for this important observation. We have now changed it to state that that SCLY “degrades Sec for the use of selenium in other proteins”

  1. Line 126 Why not use the SELENOP abbreviation as for other selenoproteins.

Thank you for the suggestion, we have changed it to SELENOP for consistency.

  1. Line 159. I might disagree that the foundation of TXNRD1 essentiality lies in "potent antioxidant cellular defense..." Isn't it more likely related to is relationship to dNTP synthesis?

We thank the reviewer for this important observation. We have added this possible explanation for the essentiality of TXNRD1 in line 578.

  1. Line 196. The sentence "Thus, like the funcion..." is pretty unclear - maybe rephrase?

Thank you for the suggestion, we have rephrased for clarity as “Thus, both supporting GPX4 activity and oxidation Sec, may be consistent with its general functionality”

  1. Line 213. If I understand how the coessentiality works, then SECISBP2L should not sort with the essential factors because it has been demonstrated to be nonessential (KO has no effect on selenoprotein production unless SECISBP2 is impaired). It would only sort as such in a SBP2 null background, correct?

In fact, it is not necessary for a protein to be essential to be part of the cluster. For example, LRP8, SECISBP2, and SCLY, all have higher scores of cell viability (figure 2b), and they still cluster with the Sec machinery based on correlation. Nonetheless, it is possible that the SECISBP2L could be part of the cluster in a SECISBP2 KO background. We have added this possibility in the manuscript in line 221.

  1. Line 239 "in" is repeated

Thank you, this has been corrected.

  1. Line 311. I don't think it is correct to call tRNASec "misacylated" by the SerRS, which implies a mistake.

Thank you, we have changed it to “aminoacylates with serine”.

Line 347.  Not only is it possible that LRP* essentiality is an artifact of cell culture, it was clearly demonstrated in Li et al. (NCB).  Along those lines, it may be important to note that SELENOP can also be taken up by other means (Megalin and pinocytosis).

We have added a sentence in discussion to point out that SELENOP can also be taken up by megalin and by pinocytosis. Line 420

Reviewer 3 Report

MAJOR COMMENTS

This manuscript uses a robust bioinformatics approach to predict a co-essentiality network for selenoproteins and partners in cancer cell lines. It is extremely well written, with thought-provoking claims that are supported by either the literature or the data presented, and it opens the doors to several novel possibilities in Se utilization in cancer. Moreover, it builds upon a previous study that had partially showed the same conclusions but without enough statistical power. The fact that the approach used cancer cell lines, however, should be stressed further, as it is possible that we may not find these factors clustering in a healthy non-proliferative-driven environment. Although functional in cancer cells, this possibility in normal cells without the same proliferative derail should be discussed.

A general comment: As SEPHS2 is also a selenoprotein that strongly appears in the co-essentiality analysis, it should be reflected in the results,discussion, abstract and even maybe in the title, that the Sec synthesis machinery is in fact sustaining the production of three selenoproteins, GPX4, TXNRD1 and SEPHS2. Currently, it is said in legend of Figure 2 and section 3.4, however this point is relevant enough to be highlighted further.

Methods to Discussion:

It is pointed that, based on the dataset used, SELENOF and TXNRD3 genes were not present, hence not used. Nevertheless, the crucial participation of SELENOF in mechanisms related to human prostate, colorrectal and breast cancer development has been established previously. This limitation of the bioinformatics approach in contrast with “wet lab” observations should be discussed.

MINOR COMMENTS

Introduction:

L. 60 - “and one Sec-specific tRNA”

L. 66: - “proteins involveD”

Results:

  • Table S1 should contain the updated selenoprotein annotation for human genes, i.e. SELENOF instead of Sep15. Also, for completeness, it should include the analysis of SCLY and LRP8 as Se-related genes in column B.
  • Legend Fig. 2: has two periods in the end.

Other: 

  • Table S1 is not listed in the Supplementary Materials.

Author Response

This manuscript uses a robust bioinformatics approach to predict a co-essentiality network for selenoproteins and partners in cancer cell lines. It is extremely well written, with thought-provoking claims that are supported by either the literature or the data presented, and it opens the doors to several novel possibilities in Se utilization in cancer. Moreover, it builds upon a previous study that had partially showed the same conclusions but without enough statistical power. The fact that the approach used cancer cell lines, however, should be stressed further, as it is possible that we may not find these factors clustering in a healthy non-proliferative-driven environment. Although functional in cancer cells, this possibility in normal cells without the same proliferative derail should be discussed.

We thank the reviewer for the critical revision of our work, and we appreciate the positive comments as well as the very relevant concerns. We have addressed all the concerns in the manuscript and have replied to each of them below.

A general comment: As SEPHS2 is also a selenoprotein that strongly appears in the co-essentiality analysis, it should be reflected in the results,discussion, abstract and even maybe in the title, that the Sec synthesis machinery is in fact sustaining the production of three selenoproteins, GPX4, TXNRD1 and SEPHS2. Currently, it is said in legend of Figure 2 and section 3.4, however this point is relevant enough to be highlighted further.

Thank you for raising this point. We totally agree that the Sec machinery is necessary for the synthesis of SEPHS2, but we consider that ultimately the Sec machinery as a whole supports the functions of GPX4 and TXNRD1. We have made changes in the text to better highlight the fact that SEPHS2 is also a selenoprotein.

Methods to Discussion:

It is pointed that, based on the dataset used, SELENOF and TXNRD3 genes were not present, hence not used. Nevertheless, the crucial participation of SELENOF in mechanisms related to human prostate, colorrectal and breast cancer development has been established previously. This limitation of the bioinformatics approach in contrast with “wet lab” observations should be discussed.

Thank you for pointing this out. We have added a comment in Methods to emphasize that these proteins have been studied experimentally, and added relevant references.

MINOR COMMENTS

Introduction:

  1. 60 - “and one Sec-specific tRNA”

Thank you, “Sec-specific” has been added.

  1. 66: - “proteins involveD”

Thank you, the typo has been corrected.

Results:

    Table S1 should contain the updated selenoprotein annotation for human genes, i.e. SELENOF instead of Sep15. Also, for completeness, it should include the analysis of SCLY and LRP8 as Se-related genes in column B.

Thank you for the suggestion. We have added a column in Table S1 with the current selenoprotein gene names, but we also kept the old names as they are currently used in the co-essentiality browser (http://coessentiality.net/), so it would be easier for readers if they search for those genes in the browser.

    Legend Fig. 2: has two periods in the end.

Thank you, this has been corrected.

Other:

    Table S1 is not listed in the Supplementary Materials.

Thank you, we have added Table S1 to the list of Supplementary Materials

Reviewer 4 Report

In their manuscript, the authors carried out an analysis of the co-essentiality network, as described in ref 8, to identify functional partners of selenoproteins and Sec machinery based on existing datasets derived from cancer cell lines. Two distinctive gene clusters are described (Sec machinery proteins; SEPHS1) with more in dept in silico analysis. This is a good example of the use of a new information and analysis to identify novel potential pathways and the authors are careful to mention shortcomings of their analysis, eg lack of experimental analyses to investigate the possible links and function of the proteins (eg Prdx6; SEPHS1…) within the clusters identified; use of cancer cell line might not be representative for function in an organism.

While the paper is clearly written overall, my main suggestion is modifying the text by moving some of the details in the result section to the discussion section. The result section contains a significant amount of detail and speculation on possible interactions and functions, what are great ideas for future work, but I feel belong in discussion section, not in the result section. If this detail is kept in result section, I suggest to investigate in more detail and add experimental data.

I believe it would improve the paper by removing some repetition and being more comprehensive, some guidelines on this:

Move line 310-323 “the biosynthesis…(figure1)” to and integrate with line 287-296.

The paragraphs from line 297- 310 “The distribution..selenium supply.” and line 334-349  “The dependency…be diminished.” would work better if they are together. They are about recycling Se and why SelenoP is no in the cluster?

I suggest that the paragraphs about non-selenoprotein synthesis factors line 324-333 and 350-362 follow each other and merge with line 153- 165 (“The glutathione peroxidase ... are needed.), line 168- 171 (“SCD is known… Sec machinery components.”)  line 174-202 (“However there…Sec pathway”) from result section, what seems to me to belong to discussion.

Rephrase and move lines 273-277 in results to integrate “Although the function of SEPHS1… with other proteins in the cluster.” into discussion (line 363-368).

Some minor remarks:

Line 130

Full description “SECISBP2” is Selenocysteine insertion sequence-binding protein 2.

Fig 1f/ Figure S1.

SCD and PRDX6 are classified as unknow, while you give a detailed description of activity/function?

Label them as a separate group?

Line 203:

“3.3. Selenium-dependent cancer cell lines”

Left Margin shifted

Line 211:

“This could be explained by the previous observation that Sec insertion might occur in the absence of SECISBP2 [35].”

How does the author explain this statement suggesting not all factors are known compared to the statement in line 134-136:” This is the first time that a functional clustering of all these Sec machinery proteins is observed. This finding supports the idea that all specialized protein components of the selenoprotein synthesis pathway had been already identified.” Similar to line 294-296:

“This is the first time that all these Sec machinery factors showed functional clustering. This finding supports the idea that all such factors had already been identified.”

Alternatively, eg. it is known from literature that SECISBP2 has complex N-terminal splicing events, with possible alternative Met start sites. Do we know which part of SECISBP2 is targeted in these cell lines to generate a KO?

I suggest these statements are rephrased more carefully, it is possible different mechanisms are possible depending on cell/tissue/physiological signalling/… and taking in account the limitations of data used in this manuscript.

Line 240

“We concluded that the differences in cell viability across lineages observed in genes in the Sec cluster (Figure 2c) were not caused by differences in gene expression (Figure 3c).”

Is it possible individual genes do not come up as causative, but differences of all/subset of genes in this pathway would?

Line 248

3.4. Selenophosphate synthetase 1 gene module”

This module is already described in reference 8, and this should be acknowledged. The possible role of SEPH1 is not discussed in ref 8, making the input with specific selenoprotein background very valued additional information.

Function/role of SEPH1 is pure speculative or is there any functionality described, then add reference.

Line 324-333.

While a detailed explanation is provided for GPX4, this is absent for TRXND1. Can this be included, move from result section?

Line 348-349

“For the same reason, the importance of endogenous SELENOP might be diminished.”

Can you change in the line of : “…SELENOP might be diminished explaining it absence from the cluster.”

Is there a possibility that different SELENOP splice variants would have different importance?

Line 360-362:

“PRDX6 is positively associated with other proteins in the cluster, suggesting that it supports their functions. The most likely possibility is that it heterodimerizes with GPX4, but it is also possible it supports TXNRD1 function.”

This is a very speculative statement. In the result section an elaborate description of possible interactions is provided and moving that here, as I suggest, would provide more evidence these interactions are possible? 

Figure S2.

Adjust the axes so numbers and names can be read.

Author Response

In their manuscript, the authors carried out an analysis of the co-essentiality network, as described in ref 8, to identify functional partners of selenoproteins and Sec machinery based on existing datasets derived from cancer cell lines. Two distinctive gene clusters are described (Sec machinery proteins; SEPHS1) with more in dept in silico analysis. This is a good example of the use of a new information and analysis to identify novel potential pathways and the authors are careful to mention shortcomings of their analysis, eg lack of experimental analyses to investigate the possible links and function of the proteins (eg Prdx6; SEPHS1…) within the clusters identified; use of cancer cell line might not be representative for function in an organism.

We thank this reviewer for the critical revision of our work and the relevant and useful suggestions. We have addressed all the concerns and replied to them below.

While the paper is clearly written overall, my main suggestion is modifying the text by moving some of the details in the result section to the discussion section. The result section contains a significant amount of detail and speculation on possible interactions and functions, what are great ideas for future work, but I feel belong in discussion section, not in the result section. If this detail is kept in result section, I suggest to investigate in more detail and add experimental data.

I believe it would improve the paper by removing some repetition and being more comprehensive, some guidelines on this:

Thank you very much for taking the time and effort to give detailed suggestions on how to improve the text.

Move line 310-323 “the biosynthesis…(figure1)” to and integrate with line 287-296.

Thank you, we have combined the two sections.

The paragraphs from line 297- 310 “The distribution..selenium supply.” and line 334-349  “The dependency…be diminished.” would work better if they are together. They are about recycling Se and why SelenoP is no in the cluster?

Thank you, we have moved the two sections together. We have also moved why SELENOP is not in the cluster after the description of SELENOP.

I suggest that the paragraphs about non-selenoprotein synthesis factors line 324-333 and 350-362 follow each other and merge with line 153- 165 (“The glutathione peroxidase ... are needed.), line 168- 171 (“SCD is known… Sec machinery components.”)  line 174-202 (“However there…Sec pathway”) from result section, what seems to me to belong to discussion.

Thank you, we have merged those parts and moved them from results to discussion.

Rephrase and move lines 273-277 in results to integrate “Although the function of SEPHS1… with other proteins in the cluster.” into discussion (line 363-368).

Thank you, we have rephrased and merged those two parts in discussion.

Some minor remarks:

Line 130

Full description “SECISBP2” is Selenocysteine insertion sequence-binding protein 2.

Thank you, we have used the full description of the protein.

Fig 1f/ Figure S1.

SCD and PRDX6 are classified as unknow, while you give a detailed description of activity/function?

Label them as a separate group?

Thank you, we believe the label separate group would not be accurate. We initially labelled them as unknown because their relationship with selenium is unknown, as indicated in the caption. We have changed now the label to “unknown Se relation” to be more specific.

Line 203:

“3.3. Selenium-dependent cancer cell lines”

Left Margin shifted

Thank you, this has been corrected.

Line 211:

“This could be explained by the previous observation that Sec insertion might occur in the absence of SECISBP2 [35].”

How does the author explain this statement suggesting not all factors are known compared to the statement in line 134-136:” This is the first time that a functional clustering of all these Sec machinery proteins is observed. This finding supports the idea that all specialized protein components of the selenoprotein synthesis pathway had been already identified.” Similar to line 294-296:

“This is the first time that all these Sec machinery factors showed functional clustering. This finding supports the idea that all such factors had already been identified.”

Alternatively, eg. it is known from literature that SECISBP2 has complex N-terminal splicing events, with possible alternative Met start sites. Do we know which part of SECISBP2 is targeted in these cell lines to generate a KO?

Thank you, we have not found the details on how the genes were targeted in the crispr knockout assays. We have however included the possibility that the SECISBP2 function could be partially retained if the SECIS binding domain was not affected.

I suggest these statements are rephrased more carefully, it is possible different mechanisms are possible depending on cell/tissue/physiological signalling/… and taking in account the limitations of data used in this manuscript.

Thank you, we completely agree that there are many nuances and unknowns in the role of SECISBP2. On our statement, we are not suggesting that there are unknown factors, we just want to point out the fact that there is some evidence of UGA readthrough in the absence of SECISBP2, which could potentially result in some level of selenoprotein synthesis in these cells, which in turn could explain why SECISBP2 seems to be less essential than other Sec machinery factors. We have now changed the sentence to point out that there is evidence of UGA readthrough instead of Sec insertion.

Line 240

“We concluded that the differences in cell viability across lineages observed in genes in the Sec cluster (Figure 2c) were not caused by differences in gene expression (Figure 3c).”

Is it possible individual genes do not come up as causative, but differences of all/subset of genes in this pathway would?

Thank you, this could be certainly a possibility. We have rephrased the sentence: “which suggests that the differences in cell viability across lineages observed in genes in the Sec cluster (Figure 2c) were not caused by changes in the expression of individual genes (Figure 3c).”

Line 248

3.4. Selenophosphate synthetase 1 gene module”

This module is already described in reference 8, and this should be acknowledged. The possible role of SEPH1 is not discussed in ref 8, making the input with specific selenoprotein background very valued additional information.

Function/role of SEPH1 is pure speculative or is there any functionality described, then add reference.

Thank you, we have acknowledged that the cluster had been described in ref 8. We also agree that discussing this cluster here has valued information.

Line 324-333.

While a detailed explanation is provided for GPX4, this is absent for TRXND1. Can this be included, move from result section?

Thank you, we have expanded the description of TXNRD1 and moved it to discussion.

Line 348-349

Can you change in the line of : “…SELENOP might be diminished explaining it absence from the cluster.”

Is there a possibility that different SELENOP splice variants would have different importance?

Thank you, we have changed the line to “It is possible that the importance of the endogenous SELENOP might have been artificially lowered on these assays due to the presence of SELENOP in the media where cells were cultured”

Is there a possibility that different SELENOP splice variants would have different importance?

It is possible that different splice variants, or truncated forms with fewer Sec residues, have different importance, however, these could not be assessed if SELENOP is present in the media.

Line 360-362:

“PRDX6 is positively associated with other proteins in the cluster, suggesting that it supports their functions. The most likely possibility is that it heterodimerizes with GPX4, but it is also possible it supports TXNRD1 function.”

This is a very speculative statement. In the result section an elaborate description of possible interactions is provided and moving that here, as I suggest, would provide more evidence these interactions are possible?

Thank you, following the suggestions of this reviewer, we have moved some parts from results to discussion. This sentence follows the detailed possible interactions.

Figure S2.

Adjust the axes so numbers and names can be read.

Thank you, we have adjusted the figure and the numbers and names are now readable.